# Maternal Methyl-Enriched Diet Increases DNMT1, HCN1, and TH Gene Expression and Suppresses Absence Seizures and Comorbid Depression in Offspring of WAG/Rij Rats

**DOI:** 10.3390/diagnostics13030398

**Published:** 2023-01-21

**Authors:** Karine Yu. Sarkisova, Ekaterina A. Fedosova, Alla B. Shatskova, Margarita M. Rudenok, Vera A. Stanishevskaya, Petr A. Slominsky

**Affiliations:** 1Institute of Higher Nervous Activity and Neurophysiology of Russian Academy of Sciences, Butlerova Str. 5A, Moscow 117485, Russia; 2National Research Center “Kurchatov Institute”—Institute of Molecular Genetics of Russian Academy of Sciences, Kurchatov Square 2, Moscow 123182, Russia

**Keywords:** absence epilepsy, animal model, depression-like comorbidity, maternal methyl-enriched diet, gene expression, WAG/Rij rat

## Abstract

The reduced expression of the HCN1 ion channel in the somatosensory cortex (SSC) and mesolimbic dopamine deficiency are thought to be associated with the genesis of spike-wave discharges (SWDs) and comorbid depression in the WAG/Rij rat model of absence epilepsy. This study aimed to investigate whether the maternal methyl-enriched diet (MED), which affects DNA methylation, can alter DNMT1, HCN1, and TH gene expression and modify absence seizures and comorbid depression in WAG/Rij offspring. WAG/Rij mothers were fed MED (choline, betaine, folic acid, vitamin B12, L-methionine, zinc) or a control diet for a week before mating, during pregnancy, and for a week after parturition. MED caused sustained suppression of SWDs and symptoms of comorbid depression in the offspring. Disease-modifying effects of MED were associated with increased expression of the DNMT1 and HCN1 genes in the SSC and hippocampus, as well as DNMT1, HCN1, and TH genes in the nucleus accumbens. No changes in gene expression were detected in the hypothalamus. The results indicate that maternal MED can suppress the genetic absence epilepsy and comorbid depression in offspring. Increased expression of the DNMT1, HCN1, and TH genes is suggested to be a molecular mechanism of this beneficial phenotypic effect.

## 1. Introduction

Absence epilepsy is a non-convulsive form of generalized epilepsies in children with a genetic cause [1]. Childhood absence epilepsy (CAE) is the most prevalent type of absence epilepsy that affects 10% to 17% of children diagnosed with new-onset epilepsy [2]. CAE is defined by the presence of typical spike-and-wave discharges (SWDs) that are associated with sudden interruption of ongoing activities with loss of consciousness (reduced responsiveness to external stimuli) [3]. Although CAE is traditionally considered a benign form of epilepsy, increasing evidence that CAE can be accompanied by neuropsychiatric comorbidities led to a revaluation of the supposed benign nature of this pathology [4]. Comorbid cognitive and attention disturbances, anxiety, and depression were found in children with CAE [5,6,7,8] and animal models [9,10,11,12]. Moreover, the pharmaco-resistance of up to 30% of treated children with CAE along with adverse effects reported for many of the currently used anti-seizure medications required further research on animal models to elaborate novel treatment options and develop new drugs with novel mechanisms of action [13,14].

The WAG/Rij rat strain is a well-validated genetic model of CAE with mild depression-like (dysthymia) comorbidity [12,14,15]. The SWDs, the main hallmark of absence epilepsy, are expressed age-dependently in WAG/Rij rats. Up to about 2 months of age, WAG/Rij rats do not have typical, well-developed epileptic seizures; only immature discharges (precursors of mature SWDs) are present on the EEG. Then, with age, immature discharges undergo three stages of “maturation” [16,17], which reflect progressive electrophysiological changes in the somatosensory cortex [17], a brain region that is related to the initiation and propagation of SWDs [18,19]. The first mature SWDs appear at about 3 months of age, and then the number and mean duration of SWDs increase [16], as well as the percentage of animals with mature SWDs. Behavioural depression-like symptoms emerge at the age of about 3 months when SWDs start to be clearly expressed. Then, with age, depressive symptoms increase, as far as the absence epileptic seizures aggravate [20].

The reduced expression of the hyperpolarization-activated cyclic-nucleotide gated subtype 1 (HCN1) ion channel and HCN1-activated current (I_h_) in the somatosensory cortex in WAG/Rij rats are thought to be associated with the genesis of SWDs [21,22,23,24]. Of interest, a rapid decline in the expression of HCN1 channels (I_h_) precedes the onset of seizures [22]. Moreover, the loss of HCN1 in HCN1-knockout rats caused spontaneous bilateral SWDs accompanied by behavioural arrest, both of which were suppressed by the anti-absence drug ethosuximide [25], indicating a causal relationship between HCN1 and absence seizures.

There is evidence that a reduced dopaminergic tone may contribute to depression-like behavioural comorbidity [20] and the genesis of SWDs [26] in genetic models of absence epilepsy. Of note, a hypofunction of the mesolimbic dopaminergic brain system, which persisted and intensified with age, was found before the absence seizure onset [26] and the appearance of comorbid depression [20]. This means that targeting the expression of HCN1 ion channels in the somatosensory cortex and the mesolimbic dopaminergic system could be a potential new approach for treating absence epilepsy and comorbid depression in the WAG/Rij model. Mechanisms underlying the downregulation of the HCN1 ion channel gene that is involved in the absence epileptogenesis are still unknown. To date, no genetic mutations in HCN1 ion channels in the absence epilepsy have been identified. De novo mutations in HCN1 were found only in early infantile epileptic encephalopathy [27]. Although pathologic phenotype in WAG/Rij rats is genetically determined, previous data have shown that the absence epilepsy and depression-like comorbidity [28], as well as HCN1 (I_h_) [23], can be altered by early postnatal environmental impacts (maternal care, neonatal maternal separation, and neonatal handling), indicating that epigenetic mechanisms might be involved. Epigenetic mechanisms regulate gene expression without altering the DNA sequence. DNA methylation is the most studied epigenetic mechanism. DNA methylation depends on the availability of dietary methyl group donors. In our previous study, we used a maternal methyl-enriched diet during the perinatal period to affect the DNA methylation in offspring and, as a consequence, to alter the development of absence seizures and comorbid depression. The results showed that a maternal methyl-enriched diet reduces the absence seizures and depression-like comorbidity in adult offspring of WAG/Rij rats. In male offspring, this beneficial effect was greater expressed than in females [29]. This study aimed to test the hypothesis that suppression of the occurrence of absence seizures and depression-like comorbidity in WAG/Rij offspring caused by a maternal methyl-enriched diet may be associated with modification of the DNA methyltransferase 1 (DNMT1), HCN1, and tyrosine hydroxylase (TH) gene expression. Selected genes were suggested as potential target genes relevant for the absence epilepsy and comorbid depression.

## 2. Materials and Methods

### 2.1. Animals

Inbred WAG/Rij rats were used as experimental subjects. WAG/Rij rats were born and raised at the Institute of Higher Nervous Activity and Neurophysiology of the Russian Academy of Sciences and represented approximately the 73rd generation from parents originally obtained from Radboud University Nijmegen (The Netherlands). All rats were kept under a natural light-dark regime (about 10 h of daytime light). Animals were housed in standard plastic cages in groups of 3–4 animals per cage. Food and tap water were available ad libitum. Experiments were performed according to the European Union Directive 2010/63/EU on the protection of animals used for scientific purposes. Animal care and use are confirmed by the institutional policies and guidelines. Experimental protocols were approved by the Ethical Committee of the IHNA (protocol № 5 of 2 December 2020) and IMG (protocol № 2 of 20 February 2019). All efforts were made to minimize the number of animals used in experiments and their suffering from experimental procedures.

### 2.2. Maternal Diet and the Offspring

At sexual maturity, 10 WAG/Rij females were placed in individual cages. Half of them had free access to the control diet (four-grain porridge, cottage cheese, and fresh eggs [29]. The other half were given a methyl-enriched diet, that is, a control diet enriched with methyl group donors and cofactors (per 1 kg of food): Choline, 5 g; Betaine, 15 g; Folic acid, 15 mg, Vitamin B12, 1.5 mg, L-methionine, 7.5 g; and Zink, 150 mg. WAG/Rij females of each group were fed with the respective diet for a week before pregnancy, during mating, during pregnancy, and for a week after parturition. After that, dams were given a standard diet. At weaning, their offspring were submitted to a standard diet with free access to food up to adulthood. The composition of methyl donors was adjusted from some previous research, showing high efficiency in modifying other genetically determined pathologies such as audiogenic seizures in Krushinsky—Molodkina strain (KM) [30] and agouti coat colour in mice [31].

### 2.3. EEG Registration, Behavioural Testing, and Gene Expression Analysis

In adulthood (7 months of age), the offspring born to mothers, fed a control or methyl-enriched diet, were randomly divided into 2 parts: one of them was subjected to EEG registration and behavioural testing (*n* = 30), and the other (*n* = 20) to gene expression analysis. To avoid “litter/mother effects”, control and experimental groups of offspring were comprised of different litters/mothers. Only male offspring were used in this study.

#### 2.3.1. EEG Registration

Stereotactic surgery was performed under chloral hydrate anesthesia (400 mg/kg, i.p.). Rats were equipped with bilateral epidural electrodes (stainless steel screws) over the frontal (AP 2 mm, L 2.5 mm) and occipital (AP −6 mm, L 4 mm) cortex. Electrodes were implanted in a small circular opening (0.8 mm in diameter) in the skull and fixed with dental acrylic. EEG registration for 3 h per day (from 16.00–19.00) was conducted in freely moving animals using the wireless 8-channels biopotential measurement system “BR8V1” based on Texas Instruments ADS1298 Analog Front-End. The EEG was recorded monopolarly, a reference electrode was placed over the cerebellum. Animals were placed in Plexiglas recording cages (15 × 30 × 26 cm) and habituated to the experimental situation for 1 h before the beginning of the recording session. SWDs were detected in the EEG as repetitive trains of sharp asymmetric large-amplitude spikes and slow waves lasting ≥1 s with amplitude at least two-fold higher than the baseline EEG signal [28]. Calculation of SWDs was performed by a ‘blinded’ unbiased expert. The severity of absence epilepsy was assessed by the number, mean duration, and index of SWDs [(total SWDs duration/recording time) × 100%]. The mean duration of SWDs was calculated as the ratio of the total duration to the number of SWDs.

#### 2.3.2. Behavioural Testing of Depression-like Symptoms

Depression-like behaviour in 7 months old offspring of WAG/Rij rats was assessed in the forced swimming and sucrose preference tests. Behavioural tests were carried out in compliance with the recommendations developed for testing epileptic animals [32].

##### Forced Swimming Test (FST)

The forced swimming test for the assessments of “behavioural despair” or depression-like behaviour originally described by Porsolt with our minor modifications was used. Only one testing procedure without a pre-test was used [12,14,28,29]. The apparatus was a cylinder (height 47 cm, inside diameter 38 cm) containing 38 cm of tap water maintained at 24 ± 1 °C. The behaviour of rats in the FST was recorded by a video camera. Rats were individually forced to swim, and the following behavioural measures were recorded for 5 min: the duration of passive swimming (immobility), the duration of the first episode of active swimming (climbing), the duration of swimming, and the number of dives and boli. Immobility was defined as no movements of the rat (floating vertically in the water, front limbs are immovable, and clasped to the breast, and nose is kept above the water surface). Climbing (‘struggling’, jumping) was defined as upward-directed strong movements of the front limbs breaking the surface of the water that resemble scratching the wall of the container. All other less vigorous movements on the water surface throughout the water tank, with the rat in a horizontal position, were defined as swimming [12].

##### Sucrose Preference Test (SPT)

Rats were kept individually and given free access to two identical bottles of tap water for habituation. On the next day, one bottle of tap water was replaced with a 2% sucrose solution. The consumption of water and sucrose was determined 24 h later. To avoid preferring the side of the bottle location, the position of the bottles (i.e., left or right) was alternated between animals. Sucrose intake (g) was measured by re-weighing the pre-weighed bottle at the end of the test. The sucrose preference was calculated according to the following equation: sucrose preference = [(sucrose intake)/(sucrose intake) + (water intake)] × 100% [12,29].

#### 2.3.3. Gene Expression Analysis

##### RNA Isolation

Tissue samples from each animal were homogenized using TRI Reagent (MRC, Cincinnati, OH, USA; Molecular Research Center, 2017). Further phase separation was performed with the addition of chloroform to the brain tissue homogenate with TRI Reagent according to the manufacturer’s recommendations. After phase separation, the aqueous phase containing RNA was separated and mixed with ethanol in a 1:1 ratio. The phenol-chloroform fraction was used for protein isolation (see below). Further isolation of total RNA was performed using RNAeasy Mini Kit (Qiagen, Solingen, Germany) and Quick-RNA Miniprep Kit™ (Zymo Research Corp., Irvine, CA, USA) according to the manufacturer’s recommendations. The RNA concentration was measured using a QuantiT RNA BR Assay Kit and a Qubit 3.0 fluorimeter (Invitrogen, Carlsbad, CA, USA). RNA quality was assessed using the Experion RNA HighSens Analysis Kit and the Experion instrument (Bio-Rad, Hercules, CA, USA).

##### Expression Analysis of Individual Candidate Genes

Gene expression analysis was performed using Real-Time reverse transcription and qPCR (RT-qPCR) with TaqMan probes. A RT-qPCR was performed on a T3 Thermocycler amplifier (T3 Thermoblock, Biometra, German, Gottingen) using the RevertAid™ H Minus Reverse Transcriptase kit (Thermo Fisher Scientific, Waltham, MA, USA) according to the manufacturer’s recommendations. A mixture of Random Hexamer Primer (Thermo Fisher Scientific, Waltham, MA, USA) and Oligo (dT) 18 primer (Thermo Fisher Scientific, Waltham, MA, USA) was used in a ratio of 3:2, respectively, for cDNA synthesis. The sequences of the primers and probes for expression analysis of the candidate genes were designed using Beacon designer 7.0 software (Premier Biosoft, San Francisco, CA, USA), and the nucleotide sequences of the Hcn1, Dnmt1, and Th genes, and the reference genes Sars and Psmd6 [33]. The specificity of primers and probes was verified using the resource Primer3 and BLAST, as well as the base IDT OligoAnalyzer 3.1 (Integrated DNA Technologies, Coralville, IA, USA). The sequences of gene-specific primers and probes are presented in Table 1.

cDNA obtained in the reverse transcription reaction was used as a template in RT-qPCR. cDNA was diluted in an aqueous solution of tRNA from Escherichia coli (100 ng/μL) to the concentration of 0.02 ng/μL before introduction into the reaction mixture. RT-qPCR was performed using the StepOnePlus™ System (Applied Biosystems, CA, USA). The reaction mixture (total volume: 30 μL) consisted of 5 μL of cDNA (0.02 ng/μL); 3 μL of PCR buffer (×10) (Synthol, Moscow, Russia); 3 μL of 25 mM MgCl2; 10 pmol of each primer (Evrogen, Moscow, Russia); 2.5 pmol of a probe (DNA synthesis, Moscow, Russia); 200 μM of each dNTP; and 1 activity unit of Taq DNA Polymerase (Synthol, Moscow, Russia). Thermal cycling was performed as follows: 50 °C, 60 s; 95 °C, 600 s; further 40 cycles of 95 °C, 20 s; and 61 °C, 50 s. Each sample was analyzed 9 times to correct for differences in sample quality and the efficiency of the reverse transcription reaction.

##### Western Blot Analysis

Total protein samples from phenol-chloroform fractions collected after the isolation of RNA with TRI Reagent (MRC, Cincinnati, OH, USA) were processed according to the manufacturer’s recommendations (Molecular Research Center, 2017). Protein concentration was measured using a Qubit Protein Assay kit and a Qubit 2.0 fluorometer (Invitrogen, Carlsbad, CA, USA). Gradient polyacrylamide gels Any kD™ Mini-PROTEAN^®^ TGX Stain-Free™ Protein Gels (Bio-Rad Laboratories, Hercules, CA, USA) were used for separating proteins by electrophoresis. Then, 75-µg aliquots of protein were loaded into each lane of the gels and the resultant protein bands were transferred to ImmunBlot^®^ Low-Fluorescence Membrane polyvinylidene difluoride (PVDF) membranes (Bio-Rad Laboratories, Hercules, CA, USA) according to the manufacturer’s recommendations using the Transfer of High Molecular Weight Proteins protocol on the Trans-Blot Turbo Transfer System (Bio-Rad Laboratories, Hercules, CA, USA) within 60 min (Bio-Rad Laboratories, Chmiel 2012). Verification experiments demonstrated that all studied proteins bound to the PVDF membranes following this protocol. The membranes were incubated with primary antibodies and nonfat dry milk as the blocking agent at 4 °C with gentle agitation overnight. The primary antibodies used in the analysis were Hcn1 (1:500, PAB28502, Abnova, Taipei City, Taiwan) and Actin B (1:10,000, PA1-16889, Thermo-Fisher Scientific, Waltham, MA, USA).

After incubation with the primary antibodies, the membranes were washed and then incubated for 1 h in the presence of the relevant affinity-purified secondary antibodies diluted 1:200,000 (P-GAR Iss, IMTEK, Moscow, Russia) at room temperature. The membranes were washed again and incubated with the chemiluminescent Super Signal West Femto Maximum Sensitivity Substrate (ThermoFisher Scientific, Waltham, MA, USA) for 5 min. Chemiluminescence was quantified using a ChemiDoc MP Imaging System and Image Lab software (Bio-Rad Laboratories, Hercules, CA, USA). Densitometric analysis was performed using the image analysis, processing, and quantification program ImageJ software (version 1.43, https://imagej.net/ij/index.html).

### 2.4. Statistical Analysis

The relative levels of the transcripts in the experimental group (*n* = 10 rats) measured by RT-qPCR were expressed relative to the control group (*n* = 10 rats) and calculated as R = 2^−ΔΔCt^ [34]. The relative protein expression levels were calculated using MS Excel 2013 software (Microsoft). Statistical data processing was performed using the “Statistica for Windows 8.0” software package (StatSoft, Inc., 2007), STATISTICA (version 8.0; www.statsoft.com), and MS Excel 2013 software (Microsoft). EEG and behavioural data were analyzed using the program “STATISTICA Release 7”. A one-way analysis of variance (ANOVA) with the Newman—Keuls post-hoc test or its non-parametric equivalent, the Kruskal—Wallis H test (one-way ANOVA by ranks), and the Mann—Whitney U test were used when appropriate.

## 3. Results

### 3.1. The Effect of Maternal Methyl-Enriched Diet on the Absence Epilepsy and Depression-like Comorbidity

The one-way ANOVA showed a significant effect of a maternal diet on the number [F(1,19) = 6.26, *p* = 0.022] and index [F(1,19) = 5.03, *p* = 0.037] of SWDs in adult offspring of WAG/Rij rats. The effect of a maternal diet on the mean duration of SWDs was not significant [F(1,19) = 0.2, *p* = 0.66]. A maternal methyl-enriched diet reduced the number of SWDs (Figure 1a), but it did not change the mean duration of SWDs (Figure 1b).

In adult (7-month-old) WAG/Rij offspring born to mothers fed a methyl-enriched diet, a decrease in the number of typical well-developed (mature) SWDs or their complete absence (≈50%) was accompanied by an increase in the number of immature SWDs usually recorded in young (2–3-month-old) WAG/Rij rats [16], indicating a slowdown in the age-dependent process of epileptogenesis.

The maternal diet had a significant effect on the immobility time [F(1,19) = 17.04, *p* < 0.001] and the first episode of active swimming or climbing [F(1,19) = 8.16, *p* = 0.01] in the forced swimming test, but did not affect the duration of swimming [F(1,19) = 2.44, *p* = 0.13]. A maternal methyl-enriched diet reduced the immobility time (Figure 2a) and increased the duration of the first episode of active swimming or climbing (Figure 2b).

The one-way ANOVA revealed a significant effect of the maternal diet on sucrose preference over water [F(1,28) = 8.83, *p* = 0.006] and sucrose intake [F(1,28) = 4.45, *p* = 0.04]. Maternal methyl-enriched diet increased sucrose preference (Figure 3a) and sucrose intake (Figure 3b), but did not affect water intake [F(1,28) = 0.64, *p* = 0.42].

### 3.2. The Effect of Maternal Methyl-Enriched Diet on the mRNA Expression Levels of DNMT1, HCN1, and TH Genes in the Somatosensory Cortex, Hippocampus, Nucleus Accumbens, and Hypothalamus

The one-way ANOVA showed a significant effect of the maternal diet on the mRNA expression levels of the DNMT1 [F(1,16) = 51.44, *p* < 0.001; *p*(U) < 0.001] and HCN1 [F(1,16) = 51.44, *p* < 0.001; *p*(U) < 0.001] genes in the somatosensory cortex. The effect of a maternal diet on the mRNA expression levels of the TH gene was not statistically significant [F(1,16) = 0.008, *p* = 0.93]. In WAG/Rij offspring born to mothers fed a methyl-enriched diet, the mRNA expression levels of DNMT1 and HCN1 genes were greater compared to WAG/Rij offspring born to mothers fed a control diet (Figure 4).

In the hippocampus, a maternal methyl-enriched diet increased the mRNA expression levels of DNMT1 [H(1,18) = 12.8, *p* < 0.01] and HCN1 [H(1,18) = 12.8, *p* < 0.01] genes (Figure 5).

In the nucleus accumbens, a maternal methyl-enriched diet increased the mRNA expression levels of DNMT1 [H(1,18) = 12.9, *p* < 0.01], HCN1 [H(1,18) = 12.9, *p* < 0.01], and TH [H(1,18) = 12.8, *p* < 0.01] genes (Figure 6).

In the hypothalamus, a maternal methyl-enriched diet had no statistically significant effect on the mRNA expression levels of DNMT1 [F(1,16) = 0.55, *p* = 0.46] and HCN1 [F(1,16) = 0.76, *p* = 0.4] genes (Figure 7).

### 3.3. The Effect of Maternal Methyl-Enriched Diet on the Protein Expression Levels of HCN1 Gene in the Somatosensory Cortex

The one-way ANOVA showed a significant effect of the maternal diet on the protein expression levels of the HCN1 gene in the somatosensory cortex [H(1,14) = 9.8, *p* < 0.01; *p*(U) < 0.01]. A maternal methyl-enriched diet increased the protein expression levels of the HCN1 gene (Figure 8).

## 4. Discussion

The results suggest that a maternal methyl-enriched diet during the perinatal period reduces the number of absence seizures and behavioural depression-like symptoms in adult offspring of WAG/Rij rats. In other words, an enhanced level of methyl donors early in development contributes to beneficial phenotypic alterations detectable in offspring long after exposure. Moreover, about 50% of 7 months old WAG/Rij offspring born to mothers fed a methyl-enriched diet had no typical well-developed (mature) SWDs. Only immature discharges, which were similar to those commonly seen in pre-symptomatic 2–3-month-old WAG/Rij rats [16], were observed. At the same time, in 100% of age-matched WAG/Rij offspring born to mothers fed a control diet mature SWDs were recorded. This means that the maternal methyl-enriched diet decelerated the age-related process of the progressive development of absence seizures in WAG/Rij offspring, indicating an antiepileptogenic effect. These results replicate, confirm, and extend the previously reported data, in which the effects of the maternal methyl-enriched diet in males and females were compared to reveal gender-dependent susceptibility to phenotypic alterations [29].

In the present study, it has been shown for the first time that increased expression levels of the DNMT1 and HCN1 genes in the somatosensory cortex and hippocampus, DNMT1, HCN1, and TH genes in the nucleus accumbens are associated with long-lasting suppression of the occurrence of absence seizures and comorbid depression in the offspring of WAG/Rij rats. Increased expression of the DNMT1 gene in the somatosensory cortex induced by a maternal methyl-enriched diet resembles the effect of early and long-term treatment with ethosuximide, a first-choice anti-absence drug [35]. In other words, a maternal methyl-enriched diet during critical periods for brain development (in utero and early postnatal stage) caused the same alterations in DNMT1 gene expression in the somatosensory cortex of offspring as long-term pharmacological anti-absence therapy. Reduced expression of the HCN1 ion channels in the somatosensory cortex in WAG/Rij rats is thought to be involved in the generation of SWDs [21,22,23,24]. There is evidence that decreased expression of the HCN1 ion channel may contribute to enhanced excitability of the somatosensory cortex and SWDs generation [24]. Of note, the effect of the maternal methyl-enriched diet on the HCN1 gene expression in the somatosensory cortex was similar to the effect of early long-term ethosuximide treatment in WAG/Rij rats [21]. Furthermore, the maternal methyl-enriched diet reduced absence seizures and comorbid depression in offspring just as much as early and long-term ethosuximide treatment [14]. Given this, increased expression of the HCN1 gene in the somatosensory cortex in WAG/Rij offspring can be associated with the beneficial phenotypic effect of the maternal methyl-enriched diet. It is possible that increased expression of the HCN1 gene may produce decreased excitability of the somatosensory cortex and, as a result, prevent the initiation and propagation of SWDs in WAG/Rij offspring. Of interest, increased expression of HCN1 in the somatosensory cortex associated with reduced absence seizures can be produced not only by maternal methyl-enriched diet but also by other early-life environmental impacts such as neonatal handling and maternal separation [23]. However, in our studies, HCN1 changed both at the mRNA and protein levels, but only at the protein level in previously reported research [23], indicating that methyl donor supplementation early in development affects HCN1 at both transcription and translation levels.

Interestingly, in the hippocampus, the maternal methyl-enriched diet caused the same alteration in the expression of DNMT1 and HCN1 genes as in the somatosensory cortex. This fact is unexpected because the hippocampus is not traditionally among the brain structures involved in the generation of typical absence seizures [36]. However, our findings indicate that the hippocampus at the molecular level is in close connection with the cortico-thalamo-cortical system. These data are in line with the existing ideas that the hippocampus plays a non-neutral role in the pathogenesis of absence seizures: it may modulate the occurrence of SWDs [37]. Moreover, during SWDs increased coupling between the somatosensory cortex and hippocampus takes place [38]. There is evidence that during spike-wave activity recorded in the cortex and thalamus, synchronization in the hippocampus increases, but this is not enough to provoke SWDs [39].

The nucleus accumbens, a key structure of the mesolimbic dopaminergic brain system, is known to be critically involved in reward, motivation, and depression-like behaviours. Mesolimbic dopamine deficiency is implicated in depression [40]. The TH gene encodes the TH, the rate-limiting enzyme in the synthesis of dopamine and is thought to play a critical role in modulating the functional activity of the dopaminergic brain system. It has been shown that TH delivered to the brain by the protein transduction domain, which passes through the blood-brain barrier, improves behavioural despair in the forced swimming and tail suspension tests, two widely used tests to screen potential antidepressants [41]. In our study, the maternal methyl-enriched diet increased the expression of the TH gene in the nucleus accumbens and rescued depression-like behaviour (reduced the duration of immobility, increased the duration of climbing and sucrose intake/preference) in adult offspring of WAG/Rij rats. Moreover, the maternal methyl-enriched diet not only increased the expression of the TH gene in the nucleus accumbens but also increased the content of dopamine, indicating an increased dopaminergic tone of the mesolimbic brain system [42]. The methyl-enriched diet that we used in our study contained components of a one-carbon cycle, the end product of which is S-adenosylmethionine (SAM), a universal donor of methyl groups not only for DNA methylation but also for the synthesis of many neurotransmitters in the brain, such as dopamine, noradrenaline, and serotonin. Insufficiency of these neurotransmitter systems in the brain is known to be associated with depressive disorders. Moreover, the reduced dopaminergic tone is also implicated to be involved in absence epileptogenesis [26]. This implies that increased expression of the TH gene in the nucleus accumbens as well as increased dopaminergic tone of the mesolimbic brain system may contribute to the suppression of absence seizures and depression-like comorbidity in the offspring of WAG/Rij rats.

Maternal methyl-enriched diet increased also the expression levels of the HCN1 gene in the hippocampus and nucleus accumbens, which may probably indicate the link between HCN1 gene expression changes in these brain structures and antidepressant-like outcomes in the offspring. There is evidence of the role of HCN channels, in particular HCN1, in depression and antidepressant effects. HCN1 channels were found to modulate the activity of dopamine neurons in the ventral tegmental area-nucleus accumbens reward pathway. However, the data are mixed and obtained on the mouse models of stress-induced depression. HCN2 channels transcriptionally and functionally were suppressed in cholinergic neurons of the nucleus accumbens shell. Increased expression of the HCN2 channels rescued depression [43]. On the other hand, global and dorsal hippocampus-restricted knockdown of HCN1 channels proteins produced an antidepressant effect [44]. HCN1 protein expression was increased in the dorsal CA1 region of the hippocampus following chronic unpredictable stress (CUS), but a reduction in HCN1 expression before CUS-induced depression provided a resilient effect [45]. There are no data concerning the expression of HCN1 channels in the hippocampus in a genetic epilepsy-associated depression-like comorbidity and its correction by antidepressant treatment. Interestingly, a reduction of HCN1, but not HCN2, was observed in a mouse model of genetic absence epilepsy. SWDs-associated transcriptional downregulation of HCN1 resulted in a reduction of I_h_ and spatial learning and memory deficits [46]. Given this, transcriptional upregulation of HCN1 in the hippocampus revealed in this study may be related to the suppression of SWDs in the offspring of WAG/Rij rats.

Although the hypothalamus is related to some aspects of depression and dopaminergic projections from the ventral tegmental area to the nucleus accumbens pass through the hypothalamus, no changes in the DNMT1 and HCN1 gene expression have been revealed. This indicates that the beneficial effect of a maternal methyl-enriched diet on the depression-like phenotype in the offspring is not related to the changes in gene expression in this brain structure.

Enhanced levels of methyl group donors early in development via a maternal methyl-enriched diet increased the expression of the DNMT1 gene in the somatosensory cortex, hippocampus, and nucleus accumbens in adult offspring of WAG/Rij rats. The DNMT1 gene encodes an enzyme that catalyzes DNA methylation (the transfer of methyl groups to the specific CpG sites in DNA). The DNMT1 methyltransferase is the maintenance methyltransferase during DNA replication. Increased DNMT1 gene expression can be assumed to modify the DNA methylation that affects many genes, some of which could be relevant for long-term disease-modifying effects of a maternal methyl-enriched diet. Our assumption is based on the prevailing view that DNA methylation is a stable and persistent epigenetic mark continuing to reproduce altered gene expression for a long time after the cessation of the impact that caused this change. We can speculate that increased expression of selected genes, potentially representing the cause or contributing to the absence epilepsy and depression-like comorbidity, is due to epigenetic modifications (changes in DNA methylation) induced by the maternal methyl-enriched diet in the early stages of offspring development. It is generally accepted that DNA methylation typically represses gene expression. However, there is evidence, indicating that methylation may also be associated with gene activation [47]. The association between DNA methylation and transcription depends on genomic context. DNA methylation represses gene expression when located in promoter regions but its effect on CpG sites in gene bodies and intergenic regions is more variable, leading to repression or activation of gene expression. DNA methylation can also directly control transcription factors binding to the regulatory region of genes, such as enhancers or transcriptional repressors, and thereby modulate the expression of genes. Furthermore, there is evidence that DNA methylation prevents binding a protein that represses transcription and, consequently, increases gene expression [48]. It is tempting to assume that an increase in DNA methylation by a maternal methyl-enriched diet during the perinatal period prevents the binding of a transcriptional repressor, such as the neuron-restrictive silencer factor (NRSF), and thereby prevents a decrease in the expression of HCN1 in the somatosensory cortex, preceding and contributing to absence epileptogenesis [22,24]. This assumption is supported by the fact that HCN1 channelopathy derives from NRSF-mediated transcriptional repression contributing to epileptogenesis, as was shown in a mouse model of temporal lobe epilepsy [49]. It cannot be excluded that methylation can directly suppress the expression of NRSF leading to a reduced transcriptional repression and, as a result, an increased expression of the HCN1gene. In other words, the increased expression of the HCN1 gene in the somatosensory cortex and possibly other studied genes may not be directly related to a change in the methylation of these genes.

An important next step will be to determine whether increased expression of candidate genes is mediated by changes in DNA methylation, and whether a maternal methyl-enriched diet elicits its beneficial phenotypic effect in offspring through changes in DNA methylation. Further, more detailed molecular studies will permit us to find out what specific mechanisms control the increase in the expression of the absence epilepsy-related genes in offspring if a methyl-enriched diet increases DNA methylation.

In this paper, we focused on the study of the expression of selected genes. However, changes in the expression of many other genes may also contribute to the beneficial phenotypic effect of the maternal methyl-enriched diet.

Emerging evidence suggests that the maternal methyl-enriched diet exerts beneficial effects on a number of pathologic phenotypes in offspring [30,31,50,51]. Although animal model studies have reported a favourable outcome, it should be noted that inadequate methyl group donor intake can harm offspring [52,53].

## 5. Conclusions

In the present study, it has been shown for the first time that an increase in the expression levels of DNMT1 and HCN1 genes in the somatosensory cortex and hippocampus, DNMT1, HCN1, and TH genes in the nucleus accumbens caused by a maternal methyl-enriched diet during the perinatal period is associated with the long-lasting (sustained) suppression of absence seizures and comorbid depression in offspring. The beneficial phenotypic effect of the maternal methyl-enriched diet during the perinatal period was similar to that of early and long-term postnatal treatment with ethosuximide, an anti-absence drug of the first choice. This fact allows us to conclude that the maternal methyl-enriched diet appears to be a new epigenetic therapeutic strategy to prevent the development of genetic absence epilepsy and comorbid depression in offspring. Epigenetic therapy based on a maternal diet is a new promising area of research. However, further in-depth studies are necessary to better understand epigenetic mechanisms by which maternal dietary supplementations during the perinatal period prevent absence epileptogenesis and comorbid depression in offspring with a genetic predisposition to absence epilepsy.

## Figures and Tables

**Figure 1 diagnostics-13-00398-f001:**
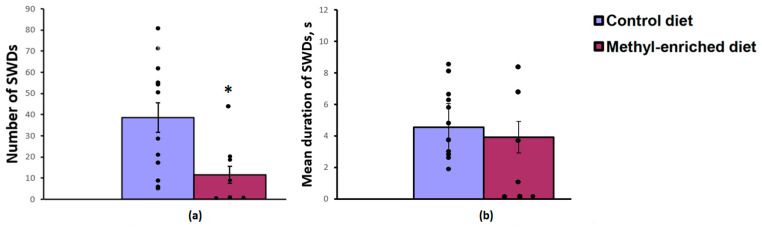
Number (**a**) and mean duration (**b**) of SWDs in adult offspring of WAG/Rij rats born to mothers fed a methyl-enriched (*n* = 8) or control diet (*n* = 13). Data are the mean ± S.E.M. The individual data points are displayed in bar graphs to show within-group variability. *—*p* < 0.05 compared with the corresponding values in the control group.

**Figure 2 diagnostics-13-00398-f002:**
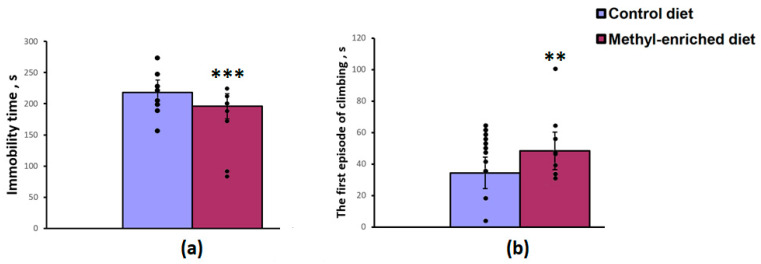
Immobility time (**a**) and duration of the first episode of active swimming or climbing (**b**) in the forced swimming test in adult offspring of WAG/Rij rats born to mothers fed a methyl-enriched (*n* = 8) or control diet (*n* = 13). Data are the mean ± S.E.M. The individual data points are displayed in bar graphs to show within-group variability. ***—*p* < 0.001, **—*p* < 0.01 compared with the corresponding values in the control group.

**Figure 3 diagnostics-13-00398-f003:**
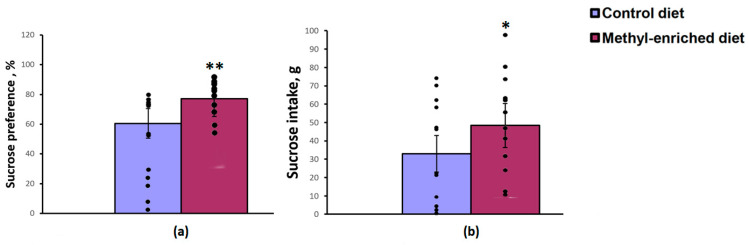
Sucrose preference (**a**) and intake (**b**) in the sucrose consumption test in adult offspring of WAG/Rij rats born to mothers fed a methyl-enriched (*n* = 16) or control diet (*n* = 14). Data are the mean ± S.E.M. The individual data points are displayed in bar graphs to show within-group variability. **—*p* < 0.01, *—*p* < 0.05 compared with the corresponding values in the control group.

**Figure 4 diagnostics-13-00398-f004:**
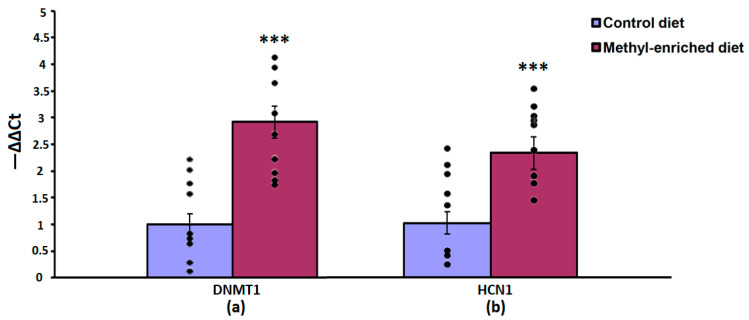
The mRNA expression levels of DNMT1 (**a**) and HCN1 (**b**) genes in the somatosensory cortex in adult offspring of WAG/Rij rats born to mothers fed a methyl-enriched or control diet. Expression of the DNMT1 and HCN1 mRNA were determined by real-time qPCR in 9 pairs of tissue samples from the control (*n* = 10 rats) and experimental (*n* = 10 rats) group of animals. Data are the mean ± S.E.M. The individual data points are displayed in bar graphs to show within-group variability. ***—*p* < 0.001 compared with the corresponding values in the control group.

**Figure 5 diagnostics-13-00398-f005:**
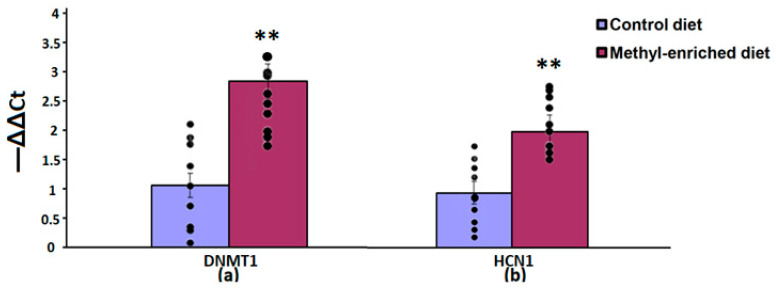
The mRNA expression levels of DNMT1 (**a**) and HCN1 (**b**) genes in the hippocampus in adult offspring of WAG/Rij rats born to mothers fed a methyl-enriched or control diet. Expression of the DNMT1 and HCN1 mRNA were determined by real-time qPCR in 9 pairs of tissue samples from the control (*n* = 10 rats) and experimental (*n* = 10 rats) group of animals. Data are the mean ± S.E.M. The individual data points are displayed in bar graphs to show within-group variability. **—*p* < 0.01 compared with the corresponding values in the control group.

**Figure 6 diagnostics-13-00398-f006:**
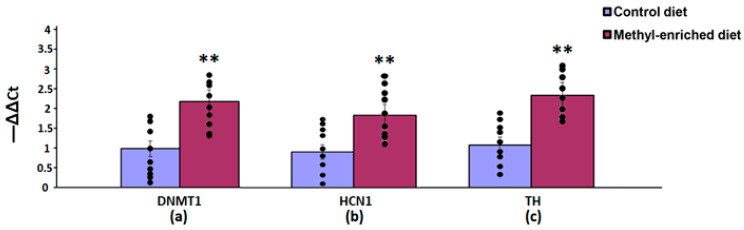
The mRNA expression levels of DNMT1 (**a**), HCN1 (**b**), and TH (**c**) genes in the nucleus accumbens in adult offspring of WAG/Rij rats born to mothers fed a methyl-enriched or control diet. Expression of the DNMT1, HCN1, and TH genes was determined by real-time qPCR in 9 pairs of tissue samples from the control (*n* = 10 rats) and experimental (*n* = 10 rats) group of animals. Data are the mean ± S.E.M. The individual data points are displayed in bar graphs to show within-group variability. **—*p* < 0.01 compared with the corresponding values in the control group.

**Figure 7 diagnostics-13-00398-f007:**
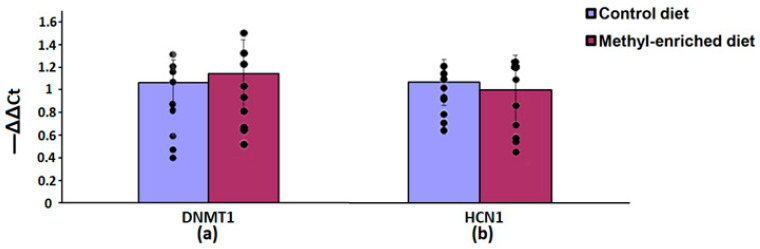
The mRNA expression levels of DNMT1 (**a**) and HCN1 (**b**) genes in the hypothalamus in adult offspring of WAG/Rij rats born to mothers fed a methyl-enriched or control diet. Expression of the DNMT1 and HCN1 mRNA were determined by real-time qPCR in 9 pairs of tissue samples from the control (*n* = 10 rats) and experimental (*n* = 10 rats) group of animals. Data are the mean ± S.E.M. The individual data points are displayed in bar graphs to show within-group variability.

**Figure 8 diagnostics-13-00398-f008:**
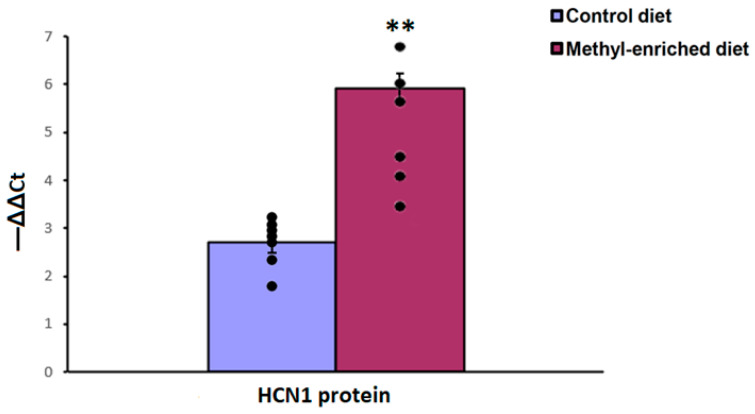
The protein expression levels of the HCN1 gene in the somatosensory cortex in adult offspring of WAG/Rij rats born to mothers fed a methyl-enriched or control diet. Expression of the DNMT1 protein was determined by western blots in 7 pairs of tissue samples from the control (*n* = 10 rats) and experimental (*n* = 10 rats) group of animals. Data are the mean ± S.E.M. The individual data points are displayed in bar graphs to show within-group variability. **—*p* < 0.001 compared with the corresponding values in the control group.

**Table 1 diagnostics-13-00398-t001:** Nucleotide sequences of gene-specific primers and probes.

Gene		Nucleotide Sequence	Fragment Length
*Sars* NM_001007606.1	Probe	5′-VIC-CGTCGCCACTCGCTGTCTGCCTTCAC-BHQ2-3′	126
	Forward Primer	5′-ACCCAGCCCTCATTCGAGAG-3′	
	Reverse Primer	5′-TCAGCTTGTTCAAGTTGTCTGC-3′	
*Psmd6* NM_196730	Probe	5′-VIC-CCTCCTTGTCACCTATCTGACAGAGGTACTCTGCTT-BHQ2-3′	117
	Forward Primer	5′-CTGGGAGAAAGTGAAATTCGAGATG-3′	
	Reverse Primer	5′-GGCCACAGTCTTATCATATGTCTTG-3′	
*Hcn1* NM_053375.1 *	Probe	5′-VIC-CGGCTCTTACTTTGGAGAAATATGCCTGCT-BHQ2-3′	135
	Forward Primer	5′-TCACCAAGTCCAGTAAAGAAATGAAG-3′	
	Reverse Primer	5′-TGTCCACCGAAAGGGAGTAAA-3′	
*Dnmt1* NM_053354.3	Probe	5′-VIC-CTACGAGGAGAACCACCAGGCAGACCA-BHQ2-3′	113
	Forward Primer	5′-GGACAGTGAGACCATGATTGAAG-3′	
	Reverse Primer	5′-CCTTGGGTTTCCGTTTAGCG-3′	
*Th* NM_012740.3	Probe	5′-VIC-CGCTGGATACGAGAGGCATAGTTC-BHQ2-3′	112
	Forward Primer	5′-TTCAATGACGCCAAGGACAAG-3′	
	Reverse Primer	5′-GTGTACGGGTCAAACTTCACAG-3′	

* Numbers in the database GenBank (Accession numbers). FAM, ROX, and VIC—fluorescent dyes; BHQ1 and BHQ2—fluorescence quenchers.

## Data Availability

The experimental data are fully presented in the manuscript. Additional information may be provided upon a reasonable request.

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
