# Peer review of "Maternal Methyl-Enriched Diet Increases DNMT1, HCN1, and TH Gene Expression and Suppresses Absence Seizures and Comorbid Depression in Offspring of WAG/Rij Rats"

_diagnostics, 2023, doi:10.3390/diagnostics13030398_

Round 1
Reviewer 1 Report
The manuscript “Maternal Methyl-Enriched Diet Increases DNMT1, HCN1, and TH Gene Expression and Suppresses Absence Seizures and Comorbid Depression in offspring of WAG/Rij rats” by Sarkisova et al is a research article which tested the effects of maternal methyl-enriched diet on gene expressions such as DNMT1, HCN1 and TH which are involved in absence seizures. The authors found that maternal methyl-enriched diet caused persistent suppression of spike-wave discharges and comorbid depression in the offspring in rats. In addition to these behavioral changes, the authors found that DNMT1 and HCN1 genes were increased in the somatosensory cortex and that DNMT1, HCN1, and TH genes were increased in hippocampus although these genes were not changed in the hypothalamus. Generally, the subject is of interest and scientifically sound and contains essential contents. This paper is also of importance for providing us the important evidence that maternal methyl-enriched diet is important for suppressing absence seizures and comorbid depression. The manuscript has been well organized and written. However, I have one concern on the paper.
In bar graphs, all the data plots should be displayed if possible. The readers can obtain more information from these data.
Author Response
Dear reviewer 1,
We would like to express our appreciation for your contribution to our manuscript. 8 graphs have been corrected in accordance with your comment.
Comment:
In bar graphs, all the data plots should be displayed if possible. The readers can obtain more information from these data.
Response:
According to your remark, 8 graphs have been corrected. We agree that the bar graphs with the data plots have become more informative, but, in our opinion, not so beautiful. A lot of black dots did not improve the appearance of the figures.
Reviewer 2 Report
Dear Authors,
I read the manuscript entitled "Maternal Methyl-Enriched Diet Increases DNMT1, HCN1, and TH Gene Expression and Suppresses Absence Seizures and Comorbid Depression in offspring of WAG/Rij rats".
The original paper, based on a preclinical study, focuses on the role of a methyl-enriched diet in suppressing epileptic seizures. The current rat study analyzed the levels of genes such as DNMT1, HCN1 and TH in male rat pups obtained from mothers fed a methyl-enhanced diet.
The paper complies with the requirements of the journal. The results are presented with the help of 8 figures, which allow the reading to be easier.
However, I have some minor questions and recommendations:
1. Right from the title you used capital letters in your writing. I would recommend that when a term is introduced in the text for the first time, the full name should be given and the abbreviation in parentheses. Later, for subsequent uses within the same manuscript, it is no longer necessary to use the full name.
2. In the Material and Methods chapter I have a number of concerns:
- You stated that the animals were provided with approximately 10 hours of light/day. In scientific research, equal cycles of 12 hours light/dark are most often used. Could you motivate the use of a 10-hour light cycle and the possible impact on the results obtained?
- What device did you use for stereotactic surgery? Please give us the name of device.
- You used chloral hydrate as an anesthetic. To my knowledge it is an anesthetic that was used for veterinary use in the past. Due to the systemic toxicity, the rather intense sedative and hypnotic effect, it is currently no longer used. How do the authors explain its use in this study?
- What impact did the anesthetic used have on the brain activity and if the results obtained by the EEG recording were modified?
- You would initially use 2 batches of animals. However, you later studied only the rat pups obtained from the 2 batches. How many male pups and how many animals were actually in the study?
3. At the end of the manuscript, you have a number of 55 bibliographic references. That's a decent number for an original item. However, there is a problem, 11 of these references belong to authors who wrote this manuscript. Self-citations are not prohibited, but I wonder how fair it is that the analysis of this article is done on a percentage of 20% also by the authors? It's like the authors self-validate your study.
4. English needs to be improved.
Author Response
Dear reviewer 2,
We would like to express our appreciation for your contribution to our manuscript. We sincerely hope that we were able to respond adequately to your comments and concerns.
Comments:
I have some minor questions and recommendations:
- Right from the title you used capital letters in your writing. I would recommend that when a term is introduced in the text for the first time, the full name should be given and the abbreviation in parentheses. Later, for subsequent uses within the same manuscript, it is no longer necessary to use the full name.
Thank you for your recommendation. The full name of genes and their abbreviation were given in the text when they appear for the first time (Introduction, page 2). We could not give the full name of genes and their abbreviation immediately after the title of the article, that is, in the abstract, since according to the requirements of the journal abstract should not contain more than 200 words (currently there are 197 words in the abstract).
- In the Material and Methods chapter I have a number of concerns:
- You stated that the animals were provided with approximately 10 hours of light/day. In scientific research, equal cycles of 12 hours light/dark are most often used. Could you motivate the use of a 10-hour light cycle and the possible impact on the results obtained?
When the studies described in this work were carried out, our animals were kept in conditions of natural daytime light (about 10 hours). Recently, our WAG/Rij rats were transferred to a new vivarium, in which ‘day’ and ‘night’ (equal cycles of 12 hours light/dark) are automatically regulated using artificial lighting close to natural. We did not find any significant difference between the impact of a 10-hour natural light or a 12-hour artificial light on the phenotype of animals or, in general, the results we obtained.
- What device did you use for stereotactic surgery? Please give us the name of device.
The stereotactic apparatus “STM-3” (Russia) was used for stereotactic surgery.
- You used chloral hydrate as an anesthetic. To my knowledge it is an anesthetic that was used for veterinary use in the past. Due to the systemic toxicity, the rather intense sedative and hypnotic effect, it is currently no longer used. How do the authors explain its use in this study?
Generally speaking, all medications may be toxic. Whether the effect will be curative, well tolerated or will be accompanied by toxic or undesirable side effects depend on the dose of the drug, the duration of exposure, the method of administration, and etc. This applies not only to anesthetics, such as chloral hydrate, but also commonly used anti-epileptic drugs, in particular such as ethosuximide, a first-choice anti-absence drug. Under certain conditions, chloral hydrate is tolerated well enough and 7 days after the surgery, no residual effects on behaviour or EEG are detected. When used correctly, there is no particular difference in the effects of inhalation or injection anesthesia using chloral hydrate or other substances.
- What impact did the anesthetic used have on the brain activity and if the results obtained by the EEG recording were modified?
In our multiple studies, we found no serious effects on brain activity or EEG recordings 7 days after i.p. injection of chloral hydrate.
- You would initially use 2 batches of animals. However, you later studied only the rat pups obtained from the 2 batches. How many male pups and how many animals were actually in the study?
We used a total of 50 male pups in this study (30 pups in the EEG and behavioural studies, and 20 pups in gene expression studies). In the sucrose preference and intake test, 30 WAG/Rij rats were used (n=14 rats born to mothers fed control diet, n=16 rats born to mothers fed methyl-enriched diet). Then, 21 rats were subjected to behavioural testing and EEG recordings (n=13 offspring from mothers fed control diet and n=8 offspring from mothers fed methyl-enriched diet). 20 WAG/Rij rats subjected to gene expression study were intact animals (to prevent the possible effect of behavioural testing or other interventions on gene expression). We have added the number of animals used in each experiment to the Material and Methods section and in the captions to the figures.
- At the end of the manuscript, you have a number of 55 bibliographic references. That's a decent number for an original item. However, there is a problem, 11 of these references belong to authors who wrote this manuscript. Self-citations are not prohibited, but I wonder how fair it is that the analysis of this article is done on a percentage of 20% also by the authors? It's like the authors self-validate your study.
Thank you for the comment. We agree. 2 references belonging to the authors of the article have been removed from the list of references.
- English needs to be improved.
We have checked the text, made corrections, if necessary, and improved English as much as possible.